# A neural crest cell isotropic-to-nematic phase transition in the developing mammalian gut

Nicolas R. Chevalier [1]✉, Yanis Ammouche[1], Anthony Gomis[1], Lucas Langlois[1], Thomas Guilbert[2], Pierre Bourdoncle[2] & Sylvie Dufour [3]

While the colonization of the embryonic gut by neural crest cells has been the subject of intense scrutiny over the past decades, we are only starting to grasp the morphogenetic transformations of the enteric nervous system happening in the fetal stage. Here, we show that enteric neural crest cell transit during fetal development from an isotropic cell network to a square grid comprised of circumferentially-oriented cell bodies and longitudinally-extending interganglionic fibers. We present ex-vivo dynamic time-lapse imaging of this isotropic-to-nematic phase transition and show that it occurs concomitantly with circular smooth muscle differentiation in all regions of the gastrointestinal tract. Using conditional mutant embryos with enteric neural crest cells depleted of β1-integrins, we show that cell-extracellular matrix anchorage is necessary for ganglia to properly reorient. We demonstrate by whole mount second harmonic generation imaging that fibrous, circularly-spun collagen I fibers are in direct contact with neural crest cells during the orientation transition, providing an ideal orientation template. We conclude that smooth-muscle associated extracellular matrix drives a critical reorientation transition of the enteric nervous system in the mammalian fetus.

[1] Laboratoire Matière et Systèmes Complexes, Université de Paris/CNRS UMR 7057, Paris, France. [2] Institut Cochin, INSERM U1016, CNRS UMR 8104, Université de Paris (UMR-S1016), Paris, France. [3] Univ Paris Est Creteil, INSERM, IMRB, Creteil, France. ✉email: nicolas.chevalier@u-paris.fr

In vertebrates, most of the ENS derives from vagal enteric neural crest cells (ENCCs). ENCCs invade the gut mesenchyme rostro-caudally early in embryogenesis[1,2]. Factors affecting the early migration of ENCCs have been the object of considerable scientific scrutiny in the past decades[3]. Current research efforts have shifted towards post-colonization morphogenetic and physiological developments of the ENS. Recent studies have for instance revealed when different subtypes of ENS neurons differentiate[4], how the ENS connects to target cells in the gut[5] and how the submucosal plexus is formed and connects to the extrinsic innervation[6]. Other investigations have unraveled how the ENS starts controlling the motility of the fetal intestine[7,8]. The morphogenesis of ENS ganglia has been the object of particular interest. Neuronal cell bodies occupy the central part of ganglia while glial cells are present both inside the ganglion, at its outer periphery, and in the interganglionic fiber tracts[9]. This arrangement as well as the characteristic size of ganglia in embryos emerge as a result of differential cell-cell and cell-extracellular matrix (ECM) adhesion[9–12]. The issue we address in this report is the genesis of the shape (morpho) of the mature murine ENS. The mammalian myenteric plexus (MP) forms a rectangular grid, where ganglia are elongated along the circumference[13–15], forming "ribs"[12]. Here, we reveal for the first time the dynamics of this orientation phase transition and, by using ENCC conditional β1-integrin mutant embryos and in-toto second harmonic generation microscopy, reveal that smooth-muscle associated ECM drives this major structural rearrangement of the ENS in the mammalian fetal gut.

## Results

**The fetal mouse enteric nervous system undergoes an orientation phase transition correlated in time with circular smooth muscle (CSM) differentiation.** We used guts obtained from mouse embryos carrying YFP or tdTomato reporter expression in the cytoplasm of all ENCCs (referred as YFP+ and Tomato+, hereafter) and their derivatives (neurons, glial cells)[16,17]. We examined the morphology of the ENS by confocal microscopy in these mice from stages E14.5 through E19.5 in the midgut (Fig.1a, b), hindgut, and cecum (Fig. S1a, b). We quantitatively assessed cell orientation with the ImageJ "OrientationJ" plugin[18], which yields the average angle of YFP+ structures (0: longitudinal, 90°: circumferential), and the orientation intensity (hereafter called coherence) at the scale of the ganglion size σ (see Materials and Methods). A high coherence indicates a high degree of anisotropy, i.e., that the cell network exhibits a distinct preferential direction. In contrast, a low coherence indicates isotropy, i.e., that the structures are only weakly or not oriented along any direction. Figure1a shows an example of orientation fields (yellow bars) in E15.5 midgut. The resulting orientation vectors are those of the dominant structure at the scale of the ganglion size, smoothing out smaller structures.

At E14.5, YFP+ cells were distributed homogeneously and isotropically in all parts of the gut (MG $n = 5$, HG $n = 4$, CC $n = 4$). In the following days, YFP+ cells became progressively oriented along the circumference of the GI tract (Fig. 1b, Fig. S1a, b), forming stripes or "ribs". ENCCs transitioned from a state of low coherence, with a slight longitudinal orientation, to a high coherence, circumferential orientation (Fig. 1c). This transition started at E14.5 in the midgut, and was almost complete by E16.5 ($n = 4$); in the hindgut it started at E15.5 ($n = 4$) and was complete by E17.5 ($n = 4$); in the cecum we found first signs of orientation at E16.5-E17.5 ($n = 4$), and complete circumferential orientation at E19.5 ($n = 4$). We compared the orientation of YFP+ (green, ENCC, Fig. 1a left) and Tuj1+ immunolabeled cells (red, ENCC-derived neurons, Fig. 1a middle) at stage E15.5. Both

labelings overlapped (Fig. 1a right), but YFP delineated cellular bodies while Tuj1 signal revealed the interganglionic fibers. While the YFP+ structure was circumferentially oriented after the nematic transition, interganglionic fibers were predominantly longitudinally oriented at all stages examined (E14.5-E16.5). The resulting structure is a roughly square grid made up of circumferentially oriented ganglia interconnected by longitudinal interganglionic fibers. This was confirmed by high-resolution confocal microscopy (using a Zeiss CSU-W1 microscope) of Tuj1+ immunolabeled whole mounts (Fig. 1d). The higher resolution of this microscope also allowed us to reveal the cellular structure of each ganglion: individual cell bodies are not circumferentially oriented, but the ganglion as the aggregate of these cell bodies is (Fig. 1d). We further monitored the dynamics of the orientation transition by time-lapse imaging E14.5 YFP+ guts in culture for up to 3 days (Fig. 1d, Supplementary Video 1). We found that clusters of YFP+ cells gradually thicken, coalesce and elongate along the circumferential direction while longitudinal connections between adjacent cell clusters are broken down (Supplementary Video 1), resulting in the emergence of rib-like structures. ENCCs transited within 30 h (Fig. 1d, bottom) from an isotropic state to an oriented cell assembly. While orientation in vivo is a robust, systematic phenomenon, we could only observe it in 10-20% of the midgut regions examined ($n = 10$ samples and $n = 40–50$ images along the midgut for each sample) after 2 days in culture (E14.5 + 2). Oriented ganglia could be found in the duodenum, jejunum, or ileum; we did not find a correlation between orientation and the rostro-caudal position along the midgut. Longer culture times resulted in the formation of large aggregates and regions devoid of ENCCs, which indicated a deviation from physiological development.

Because CSM is also circumferentially oriented, we investigated whether the orientation of ENCCs and their derivatives could be driven by CSM. We determined the time of CSM differentiation (CSMD) in the different parts of the GI tract by whole-mount immunohistochemistry for smooth muscle α-actin (Fig. 2a) and by measuring the time of first appearance of contractile waves (motility, Fig. 2b); we have previously demonstrated in the chicken that the emergence is synchronous with CSMD[19]. We found that circumferentially oriented α-actin fibers appear at E13.5 in the jejunum, at E14.5 in the hindgut and at E15.5-E16.5 in the cecum (Fig. 2a,c); the onset of contractions was synchronous with the differentiation of CSMD in all regions of the GI tract (Fig. 2b), further confirming this chronology. The time lag between CSM differentiation in the mid- and hindgut and the cecum is consistent with observations in the chicken embryo[19]. In every part of the murine gut, orientation of YFP+ cells thus starts 1 day and finishes 3 days after CSMD. This constant time lag suggests that CSM or CSM-associated ECM acts as a cue to guide ENCC circumferential orientation.

**ENCC adhesion through β1-integrins is critical for the orientation transition.** ENCCs express various integrins that control their adhesion to gut ECM[20]. We previously showed that NCC-specific conditional invalidation of the *Itgb1* gene encoding the β1-integrin subunit (*Itgb1*-cKO) produces aganglionosis of the distal colon[21] due to defective ENCC-ECM adhesive interactions, altered migration, and exacerbated intercellular adhesion[11,17]. At E16.5, the ENS present in the colonized part of the conditional *Itgb1* mutant guts displays a distorted pattern (Fig. 3b) compared to the typical stripe pattern of the ENS in control embryos (Fig. 3a). We quantitatively assessed the orientation of ENCC network in control ($n = 4$) and *Itgb1*-cKO ($n = 3$) E16.5 midguts (Fig. 3c). Control ENCC networks exhibited a circumferential orientation (80.9 ± 2.6°, all results expressed as mean ± SD) with a

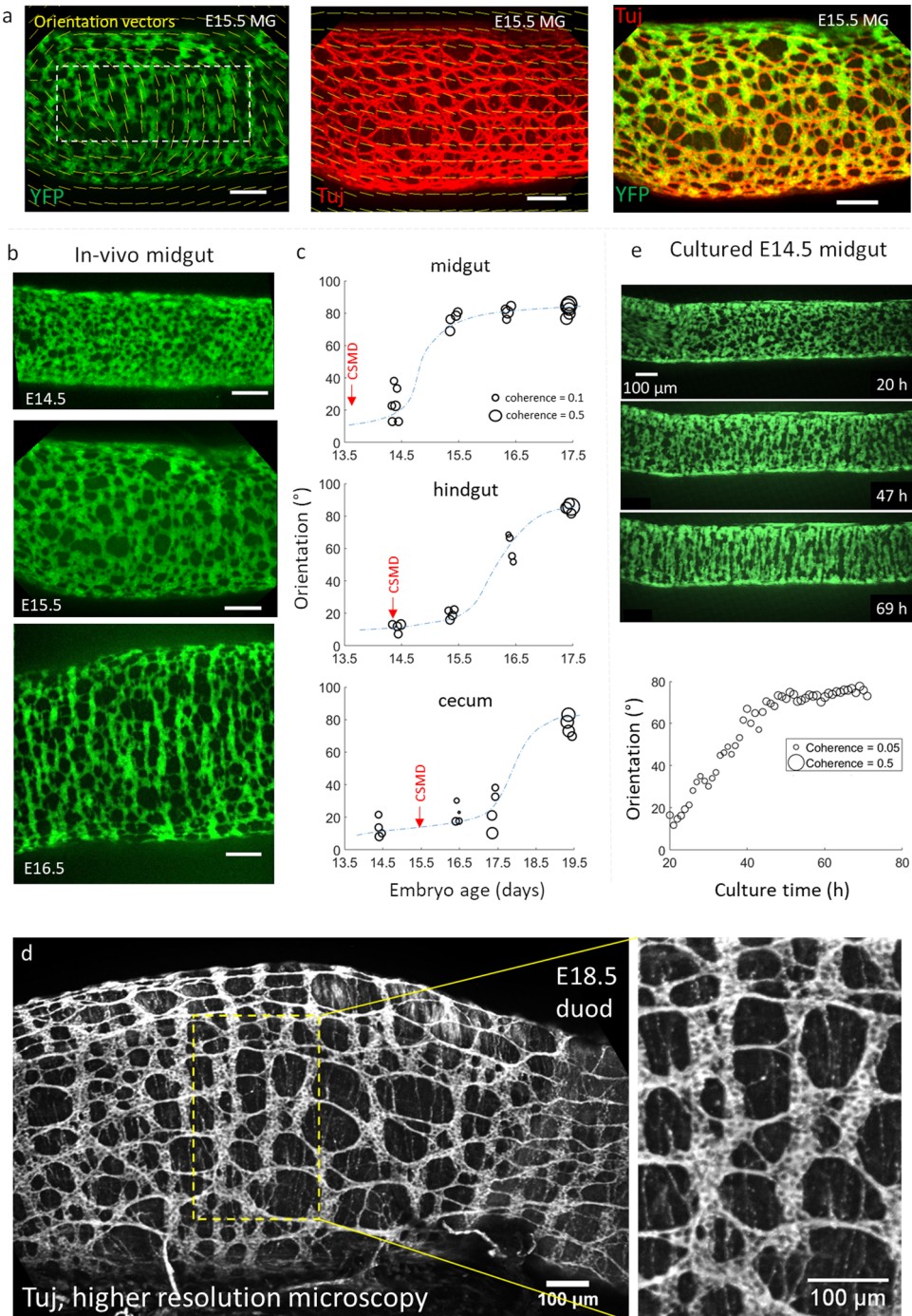

**Fig. 1 Morphogenesis of the murine fetal ENS. a** OrientationJ orientation field (yellow vectors) and coherence (α vector length) analysis results for YFP+ cells (left), βIII-tubulin immunolabeling (neurons, interganglionic fibers, middle), and overlap (right). Dashed rectangle: ROI for averaging of orientation & coherence. **b** Orientation of YFP+ cells in the midgut from E14.5 to E16.5. **c** Evolution of the orientation and coherence in the midgut, hindgut, and cecum. Each dot is a different sample (embryo). The dashed blue sigmoid is a guide to the eye, the red arrow indicates the time of circular smooth muscle differentiation (CSMD), see Fig. 2. **d** High-resolution confocal microscopy reveals cell bodies within individual ganglia; βIII-tubulin immunolabeling, max z-projection of confocal stack (**e**) Still shots from Supplementary Video 1 showing progressive circumferential orientation of YFP+ cells in cultured E14.5 jejunum. Bottom graph: orientation and coherence evolution extracted from Supplementary Video 1.

high coherence (0.48 ± 0.05). In contrast, the *Itgb1*-cKO network revealed more longitudinal and scattered orientations (61.7 ± 10.6°), with a lower coherence (0.30 ± 0.08). These differences were even more conspicuous at stage P1 and P14 (see Fig. 2 in Breau et al.[21]). This indicates that β1-integrins mediated ENCC adhesions are required for the proper orientation transition of the ENS.

**ENCC are in contact with circular collagen fibers during the orientation transition**. We next investigated the nature of the ECM associated with enteric neurons and glia. We used label-free second harmonic generation (SHG) microscopy to reveal the 3D spatial orientation of fibrous collagen relative to that of the Tomato+ ENCC network; this information cannot be gathered from antibody labeling on 2D slices. In the context of the

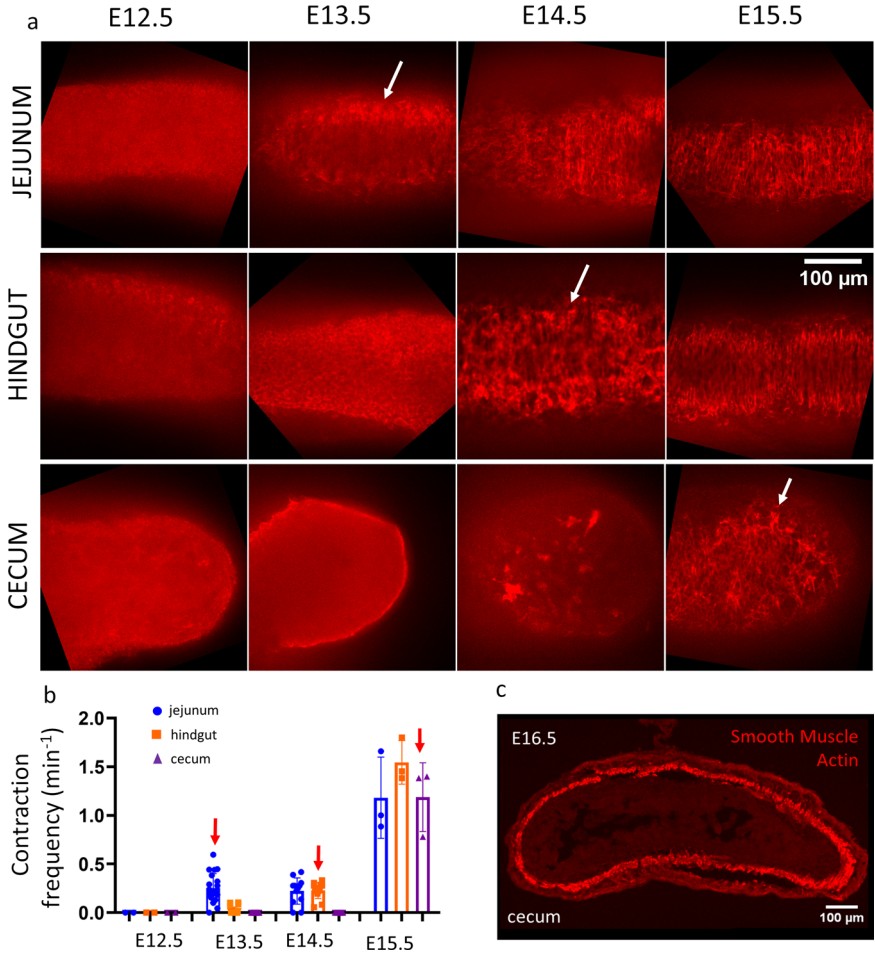

**Fig. 2 Circular smooth muscle differentiation (CSMD) takes place at different times in different parts of the lower GI tract. a** Whole-mount immunohistochemistry for smooth muscle α-actin. The white arrows indicate the first appearance of circumferential muscle fibers. The scale is indicated at the right and is the same for all images. **b** Average frequency of peristaltic contractions in the mouse midgut, hindgut and cecum at E12.5 ($n = 2$), E13.5 ($n = 20$), E14.5 ($n = 12$), and E15.5 ($n = 3$), in cycle per minute (cpm). Each data point corresponds to a different embryo (gut). The red arrows indicate the first appearance of contractile waves (motility). Note that in the cecum at E15.5 the contractions were circumferential (i.e., local diameter reduction) only at the base of the cecum, but appeared as "bulk" contractions in the remainder of the cecum. They only became circumferential in the whole cecum as from E16.5. **c** Thin section immunostaining in E16.5 cecum showing full differentiation of the smooth muscle layer at this stage.

intestine, the SHG signal is proportional to the density of collagen I fibers (Materials and Methods).

Collagen I is a permissive α1β1 integrin-mediated adhesion substrate for NCCs[22,23]. We found that circularly oriented collagen fibers were present in the plane and immediately below the myenteric plexus (MP) at the start (E14.5, Fig. 4a, Supplementary Video 2) and after ENCC orientation was complete (E17.5, Fig. 4b, Supplementary Videos 3, 4). This fiber layer was very thin (a few μm) at E14.5 (Fig. 4a), co-located to a large extent with the ENCCs (Fig. 4a, $z = 80$ μm, merge), but was situated more towards the inside of the intestine, at the boundary between the CSM and the myenteric plexus. SHG microscopy also revealed the collagen-rich serosa (Fig. 4a, $z = 80$ μm). At E17.5, the collagen fiber layer became much thicker (~50 μm, Fig. 4b). The intensity of the SHG signal just below the myenteric plexus increased 5-fold between E14.5 and E17.5, from $25 \pm 5$ pixel units ($n = 7$ stacks from $n = 4$ guts) to $128 \pm 34$ ($n = 9$ stacks from $n = 4$ guts). At E17.5, collagen fibers not directly in contact with the ENS plexus, in deeper layers of the gut (the submucosa, where most of the collagen is located[24]), had various orientations; in the duodenum and midgut we found circular, oblique (visible on the left side of Fig. 4b, $z = 10$ μm), or bi-diagonal (with fibers at ±45°) orientations. The collagen

distribution in the MP, SMC, and submucosa we report is consistent with immunostaining of collagen I on E18.5 duodenum thin-sections presented by other investigators[24]. We found similar results in the E17.5 hindgut (Fig. S2). Interestingly, the outer part (closer to the serosa, Fig. S2, $z = 0$ μm) of the MP in E17.5 hindgut was remarkably less circularly oriented than its inner part (close to the CSM, Fig. S2, $z = 10$ μm). This indicates that reorientation of the ENS cells is a very local phenomenon, which only occurred in direct contact with the circular ECM fibers. Intense longitudinally oriented collagen fibers were located in the submucosa of the E17.5 hindgut ($n = 4/4$, Fig. S2, $z = 30$ μm). ENCCs in the hindgut were not in direct contact with these deeper longitudinal fibers (see gap indicated by arrowhead in Fig. S2, $z = 120$ μm). We finally compared the fiber pattern revealed by SHG microscopy with the one obtained by collagen I whole-mount antibody immunohistochemistry on E17.5 gut (Fig. S3, Supplementary Video 5). We found that collagen fibers are wound circularly at the level of the myenteric plexus, as found by SHG in Fig. 4b. We could not however reveal collagen deeper in the mucosa, either because the antibody did not penetrate the deeper tissue layers, or because we used a spinning disk microscope that has lower penetration depth than the two-photon SHG method. Antibody staining further labeled

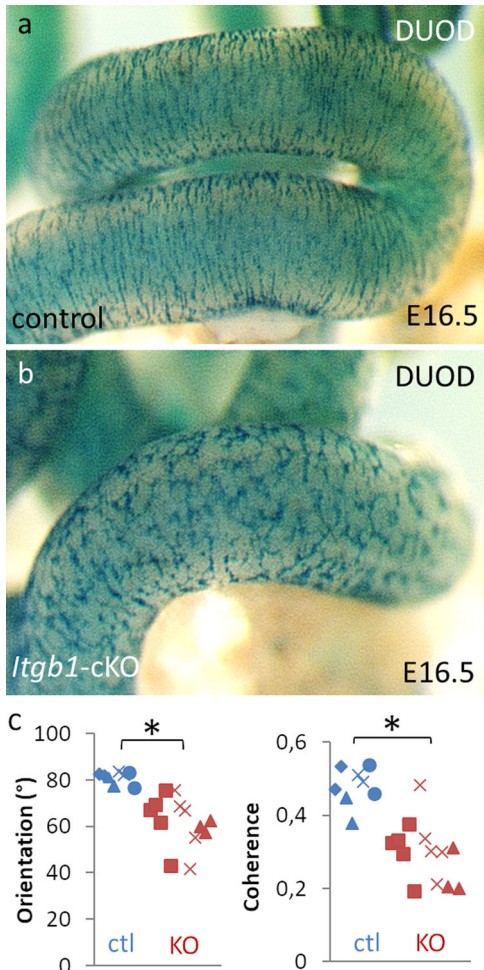

**Fig. 3 Conditional β1 integrin knock-out in ENCCs perturbs their circumferential reorientation.** Whole-mount X-Gal staining of (**a**) control and (**b**) Itgb1-cKO ENS in E16.5 gut. (**c**) Orientation and coherence of control and Itgb1-cKO. Each symbol corresponds to a different region analyzed along the midgut (2 to 5 analyzed per embryo); different symbol shapes correspond to different embryos ($n = 4$ control, $n = 3$ Itgb1-cKO). *: statistically significant difference, $p < 0.05$, two-tailed Mann–Whitney test.

individual collagen-rich cells below the myenteric plexus, that we did not observe by SHG microscopy.

## Discussion

We have demonstrated that CSM differentiation and ENS circumferential orientation were temporally tightly correlated, the first preceding the latter by exactly 1 day in all regions of the lower gastro-intestinal tract (Figs. 1, 2). We captured the dynamics of this orientation transition (Fig. 1e, Supplementary Video 1) by ex vivo time-lapse imaging. In analogy to the physics of liquid crystals, the orientation transition can be assimilated to an isotropic-to-nematic transition, where the isotropic phase does not exhibit a preferential direction, whereas the nematic phase is ordered along one direction (the circumference of the gut). We showed that Itgb1-cKO ENCCs depleted of β1-integrins display a defective circumferential re-orientation (Fig. 3). While the orientation transition in Itgb1-cKO ENCC is perturbed, it is not completely abolished, suggesting that other types of integrin receptors besides β1 may be involved, or a correct balance between integrins and cadherins[11]. We revealed circularly oriented collagen I fibers at the MP–CSM interface by second harmonic generation microscopy (SHG, Fig. 4); the density of

these fibers increased during the physiological orientation of the plexus between E14.5 and E17.5. Collagen fibers could provide an ideal template for ENCC orientation, although other ECM molecules might be involved. We conclude that the reorientation of ganglia along the circumference of the intestine is driven by smooth-muscle associated ECM.

Candidate ECM molecules that could serve as a template for ENCC orientation via β1-integrins should fulfill the following criteria: (1) be present in the embryonic gut at the level of the myenteric plexus at the stages investigated (E14.5-E19.5), (2) be permissive to ENCC adhesion, (3) present a circumferential organization, that is be of fibrous nature, (4) be associated/produced by the smooth muscle, because we showed that there is a kinetic correlation between the appearance of smooth muscle and the reorientation of ENCCs. Collagen I, III, and V are the main collagen molecules present in the gut, representing respectively 68, 20, and 12% of collagens in the human intestine[25]. Collagen III and V are generally found associated with collagen I fibers[26,27] and for the sake of this discussion we will therefore consider these three fibrous collagen types as one entity. Collagen I have been shown to be permissive to NCC migration[22,23]; in particular, the orientation and translocation of NCC along (parallel to) collagen I fibers has been reported by Davis[28]. Importantly, collagen I, III, and V have been shown to be produced by fetal human intestinal smooth muscle cells[29]. We have shown here by second-harmonic generation microscopy that collagen I is present at the right location (myenteric plexus), stage (E14.5-E17.5) and with the proper circumferential orientation. In contrast, collagen IV forms a sheet-like structure which, although permissive to migration[23], is not present at the level of the myenteric plexus in the hindgut when the orientation transition occurs[24]. Type VI collagen is mostly localized at the basal membrane of the epithelium[30], but is also secreted by ENCCs. It however was found to have an inhibitory effect on migration, with overexpression triggering a Hirschsprung type phenotype[31]. Collagen XVIII is only secreted by ENCCs at the colonization wavefront[32]. We can therefore conclude that, among collagens, collagen I fibers (and associated type III and V) fulfills the criteria of an ideal template ECM for ENCC orientation. We cannot however exclude that other, non-collagenous ECM molecules that support ENCC migration, like fibronectin[31], could be produced in an oriented way by the CSM and play a role in driving the orientation transition of ENCCs.

Does the orientation transition occur in other species? Whole-mount staining of peripherin and Phox2b in transparized human embryonic midgut at 9 and 13 weeks of development[32] (see also http://transparent-human-embryo.com) shows that the stripe pattern of the ENS is present in humans as well: peripherin filaments run parallel to the longitudinal axis whereas Phox2b positive ganglia are oriented circumferentially. The ENS orientation transition in humans probably occurs between weeks 7 and 9 of development, just after CSM differentiation[33,34]. In contrast to the mouse colon, the MP of the human colon has a hexagonal honeycomb geometry[35–37] which resembles more closely that of the chicken hindgut.

When CSMD is inhibited in the chicken midgut, the regular plexus geometry is lost and ENCCs chaotically invade regions closer to the epithelium[38]. Enteric ganglia in the chicken do not however reorient like in mammals, in spite of the presence of circular collagen I fibers associated with the CSM[22]. The ENS of the chicken takes on a hexagonal honeycomb geometry shortly after ENCC colonization, and this geometry does not change later in development. Differences in the balance of forces pushing for orientation (ECM-ganglion interactions) and the ones opposing it (cell-cell adhesion and interganglionic fiber tension) are the likely reason for these species-specific variations in the final ENS morphology. Interganglionic fibers in chicken are noticeably

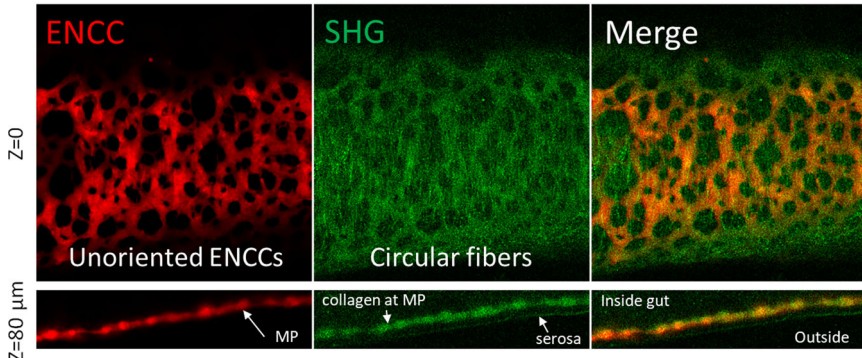

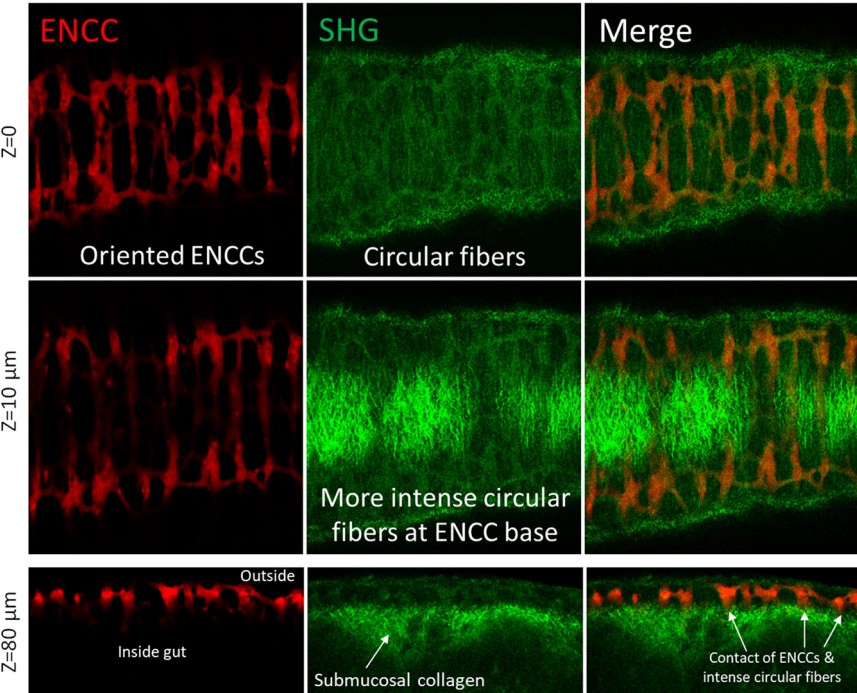

**Fig. 4 ENCC and collagen imaging by second-harmonic generation microscopy. a** E14.5 duodenum, at the start of ENCC reorientation, plane selected from z-stack (Supplementary Video 2) at the level of the myenteric plexus (MP). Collagen fibers are circumferentially oriented. $Z = 80\ \mu m$ is a longitudinal optical section of one border of the gut. The collagen is concentrated at the serosa and at the MP. **b** E17.5 duodenum, after ENCC reorientation. Circumferentially oriented collagen fibers are situated at ($z = 0\ \mu m$) and just below the MP ($z = 10\ \mu m$). The strong collagen signal extends further in the submucosa ($z = 80\ \mu m$, longitudinal optical section of one border of the gut), where it had various orientations (circular, longitudinal or oblique) depending on the sample and sample region examined.

thicker than in mice and longitudinally oriented; the pulling force they exert on ganglia may offset the tendency of ganglia to orient circumferentially with the underlying ECM. An interesting question is the implication of these differences of ENS geometry on the physiology of the intestine. Neurons in the ENS are known to be mechanosensitive[39] and we expect that the mechanical stresses (due to the presence of food bolus or spontaneous muscle contractions) experienced by circumferential vs hexagonal ENS lattices will be significantly different.

We shall conclude by advancing the hypothesis that internal mechanical forces are directing the orientation transition *via* the ECM. It is known that the more important proliferation of epithelial compared to mesenchymal cells induces a circumferential internal mechanical stress[40] that is present at least as from E12.5 in the murine intestine. This internal stress is responsible for the circular orientation of the CSM in the chicken[40]. Since static

stresses are well-known to orient collagen fibers[41], it is not hard to imagine that circumferential internal stress could orient the CSM-associated ECM, which in turn orients the myenteric plexus by the mechanism outlined here. This points to the potentially long-ranging implications of small internal stresses in organ morphogenesis.

As in central nervous system development[42], our research stresses the key role played by interactions between enteric neural crest cells and the extra-cellular matrix in driving major structural rearrangements of the enteric nervous system during mammalian fetal development.

## Methods
**Ethics.** All procedures presented in this manuscript have been approved by the ethics committee ComEth Anses/ENVA/UPEC, APAFiS request number

2019061218208141 (#210725). All experiments were performed in accordance with the ethics guidelines of the INSERM and CNRS.

**Specimens, samples.** Mouse models used in this study are the following. The Cre reporter mice Gt(ROSA)26Sortm1(EYFP)Cos[43], and B6.Cg-Gt(ROSA)26Sortm9 (CAG-tdTomato)Hze/J[44] were refereed as YFP[fl/fl] and Tomato[fl/fl], respectively. A transgenic mouse line in which the transgene is under the control of the 3-kb fragment of the human tissue plasminogen activator (Ht-PA) promoter Tg(PLAT-cre)116Sdu[16], was referred as Ht-PA::Cre. YFP[fl/fl] or Tomato[fl/fl] were cross with Ht-PA::Cre to generate embryos carrying the reporter protein in migrating NCCs and their derivatives.

The mice carrying floxed *Itgb1* allele Itgb1tm1[45] and the mice carrying one *Itgb1* null allele Itgb1tm2[46] were referred as beta1[fl/fl] and beta1[neo/+], respectively. Ht-PA::Cre were crossed with beta1[neo/+] to generate Ht-PA::Cre; beta1[neo/+], which were then crossed with beta1[fl/fl] to generate embryos with Ht-PA::Cre; beta1[neo/fl] or Ht-PA::Cre; beta1[+/fl] genotype as previously described[17]. Ht-PA::Cre; beta1[fl/+] embryos that are heterozygous for *Itgb1* in migrating NCCs were referred to as controls because *Itgb1* heterozygosity has no effect on mice phenotype[47]. Ht-PA::Cre; beta1[neo/fl] embryos that carry the conditional mutation for *Itgb1* in migrating NCCs were referred to as *Itgb1*-cKO. In both controls and *Itgb1*-cKOs, Cre recombinase-mediated deletion of the beta1[fl] allele leads to the expression of beta-galactosidase under the control of the *Itgb1* promoter[46]. This allowed us to visualize the NCCs and their derivatives including ENS using Xgal staining. The day of the vaginal plug was considered E0.5.

**Statistics and reproducibility.** All sample numbers indicated in this report correspond to different embryos (guts, biological replicates). Information was collected from 1 or 2 different litters per age. All experiments involving culture were technically replicated at least twice, i.e. they were performed on two different days with fresh samples following the same procedure each time. A minimum of $n = 4$ samples constitutes each group presented in this report. Statistical analysis was performed with the two-tailed Mann-Whitney test and was considered significant at $p < 0.05$.

**Organotypic culture.** E14.5 mice guts were cultured in 1 mL DMEM with 1% penicillin-streptomycin in 35 mm diameter Greiner culture dishes whose bottom was covered with a thin layer of Sylgard. Guts were pinned with 100 μm steel pins (Euronexia) at their extremities (stomach and distal hindgut) and cultured for 72 h in a humidified incubator at 37 °C, in a 5% $CO_2$ atmosphere. For live time-lapse confocal imaging, the gut was embedded at the surface of a 2% type-VII agarose gel to reduce sample displacement. Live confocal imaging was performed on a DSD2 confocal microscope at x10 magnification, at a frequency of 1 image/10 min, in a humidified 5% $CO_2$ atmosphere. Z-stacks were collected and the maximum projected. Jitter motion of the gut (Supplementary Video 1) is due to spontaneous CSM contractions that propagate along the sample length. These contractions occur at a smaller time scale (on the order of 1–10 s) than the gradual orientation of the ENS. Fixed-time imaging of guts was performed on an Olympus spinning disk confocal microscope with GFP excitation/emission filters (488/512 nm). Given the small proportion of mouse gut samples (10–20%) that exhibited re-orientation in culture and the limited availability of our confocal microscope for prolonged time-lapse imaging, we were only able to produce one video (Supplementary Video 1) in which we could track the re-orientation of YFP+ cells along the circumference.

**Monitoring of gut motility.** Spontaneous motility of E12.5-E14.5 mice gut was monitored by placing them in 1 mL of DMEM in individual dishes at 37 °C and time-lapse imaging with a binocular (MZ series, Leica) and camera (Stingray) at a 1 Hz frequency for at least 5 min. Space–time diagrams of motility[19] in different regions were extracted with the imageJ "Reslice" function.

**Immunohistochemistry.** For IHC of frozen sections, guts were fixed for 1 h in a 4% PFA in PBS solution, washed in PBS, then let overnight in 30% sucrose in water solutions, and embedded the next day in OCT compound (VWR) on dry ice. Thin (14 μm) slices were cut at −20 °C in a Leica cryotome and deposited on Thermofrost glass slides. After rehydration, the slides were blocked for 15 min in a 1% BSA and 0.1% triton in PBS solution, the slides were then incubated overnight in anti α-smooth muscle actin antibody (Abcam, ref5694, dilution 1:2000) and/or anti βIII-tubulin antibody (Tuj-1, Abcam, ref14545, dilution 1:1000) solution composed of 1% BSA in PBS; the following day, after washing, CY3- and A488-conjugated secondary antibodies (ThermoFisher, dilution 1:400 in PBS) were applied for 2 h. The slides were washed in PBS, and immediately imaged in a layer of PBS with a confocal microscope (low-resolution Olympus system or high-resolution Zeiss CSU-W1). For whole-mount Tuj1 staining, after PFA-fixing, the gut was blocked and permeated in 1% BSA and 0.1% triton in PBS overnight, immersed in anti βIII-tubulin antibody, as described above, for one day, washed and immersed in Alexa488 secondary antibody for at least 4 h. For whole-mount collagen I (Sigma SAB1402151) IHC, primary antibody was incubated at 1:100 for 3 days, and

revealed with Alexa647 secondary antibody (1:400, overnight). The detection of beta-galactosidase activity was revealed by whole-mount X-gal staining as described in Breau et al.[21].

**Harmonic and fluorescence imaging.** Demesenterized guts were embedded at the surface of a 3% agarose gel immersed in PBS. Images were acquired with an upright Leica SP8 DIVE (Leica microsystems Gmbh, Wetzlar, Germany) coupled with a Coherent Discovery (Coherent Inc., Santa Clara, CA, USA) pulsed laser and controlled by LAX imaging software. Both 552 nm continuous laser and 1040 nm pulsed laser were used to acquire tdTomato fluorescence and second harmonic generation, respectively, through a ×25 objective in PBS immersion. Because of the intrinsic high-penetration depth (~100 μm) of two-photon microscopy techniques, SHG can be used to image the orientation of collagen fibers in the relevant optical sections, without the need to actually perform physical sectioning of the sample. In contrast, antibody staining of collagen is most often performed on tissue sections, on which it is very difficult to assess the overall orientation of cellular and extracellular matrix components (e.g. the orientation of ENCCs wound around a cylinder as in Fig. 1 cannot be detected on physical sections). SHG allows for quantitative evaluation of collagen fiber density. SHG is sensitive to collagen type I and type II[48], but since only type I is found in the gut (where it is the main collagen type expressed[25]), we image here the collagen I distribution. Comparative images of collagen I distribution by immunostaining and of SHG generation in liver samples have for example shown almost complete overlap[49]. We note that SHG is insensitive to non-fibrillar collagen-like type IV; type III does not generate a measurable SHG signal[48].

**Image analysis.** Orientation of the ENS in the different segments of the mouse was measured with the ImageJ "OrientationJ" developed by the Biomedical Imaging Group at EPFL[18]. This plugin computes a vector describing the mean orientation of structures of size σ in a region-of-interest (ROI). We used the Gaussian gradient method and a size σ close to that of the ganglia; choosing too small σ traces the contour of individual ganglia and does not give meaningful information about their average orientation. The direction of the vector represents the orientation of the subimage while its length (proportional to the coherence) quantifies the degree of anisotropy. The sides of the gut present an artefactual longitudinal orientation induced by 2D z-stack projection of the 3D cylindrical gut tube. We therefore excluded the sides from the region-of-interest (Fig. 1a dashed white rectangle) for the orientation analysis. We averaged the coherence and direction of all structures present within the ROI.

**Reporting summary.** Further information on research design is available in the Nature Research Reporting Summary linked to this article.

## Data availability

All data generated or analyzed during this study are included in the manuscript and supporting files (Supplementary Material & Supplementary Data 1). Any remaining information can be obtained from the corresponding author upon reasonable request.

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

## Acknowledgements

The authors thank Benoit Sorre for granting access to the DSD2 confocal microscope, Melina Durande and François Graner for advice on orientation measurements, Don Newgreen for feedback on our manuscript. We thank the Mouse Facility staff of the Institut Mondor de Recherche Biomédicale. We thank the Imagoseine facility for use of the Zeiss CSU-W1 spinning disk microscope. This research was funded by the CNRS Défi Mecanobiologie "MECHENSDEV" grant, by the IdEx Université de Paris ANR-18-IDEX-0001 grant, by the Agence Nationale de la Recherche ANR GASTROMOVE-ANR-19-CE30-0016-01 grant and by the Institut National Pour la Santé de la Recherche Médicale (INSERM).

## Author contributions

N.R.C. performed experiments, conceptualized & supervised the research, produced figures, wrote the paper with support from S.D. and T.G.; Y.A. performed quantitative analysis and produced figures; A.G. and L.L. performed experiments; T.G. and P.B. performed experiments; S.D. genetically engineered the mouse strains, provided samples and critically revised the paper.

## Competing interests

The authors declare no competing interests.
