## [Peer Review File · Communications Biology]

Reviewers' Comments:

Reviewer #1:

Remarks to the Author:

There has been much interest in the colonisation of the embryonic gut by neural crest cells.

In this study by Chevalier et the authors suggest they have revealed the structural arrangement of the myenteric plexus is a result of a dynamic neural crest phase transition (isotropic-to-nematic) that occurs in the fetal gut. The authors suggest that the orientation parameter is critically dependent upon circular smooth muscle associated collagen microfibrils that enteric neural crest cells anchor to via integrin mechanoreceptors.

The authors used mouse embryos that expressed YFP or td Tomato in the cytoplasm of all enteric neural crest cells and then used confocal microscopy to follow the morphology of the enteric nervous system from E14.5 to E19.5 in the mid gut, hindgut and cecum. The authors assessed the relative orientation intensity or coherence.

The paper is very descriptive and lacks quantitation in areas. The authors use a knockout mouse (Itgb1-CKO) and state they "...revealed more longitudinal and scattered orientations, with lower coherence (~.35)..." This is an imprecise description. And, measurements should be precise with accurate S.E.Ms, not ~0.35.

The notion that Integrins are important for enteric neural crest migration is not new. Work (cited in this article) by Breau MA et al. (2006) Development 133: 1725-34 has demonstrated the lack of B1 integrins leads to a Hirschsprung's like phenotype.

The authors use the term "integrin mechanoreceptors". Why? What is the evidence they perform this role in this environment?

The manuscript is quite descriptive. I appreciate that the movement of the ENCC is not easy to describe, but phases like "...YFP+ cells gradually thicken, coalesce and elongate along the circumferential direction while longitudinal connections between adjacent cell clusters are broken down, resulting in the emergence of rib-like structures" are not very helpful.

In many places, there are a low number of animals used (often N=1, N=2 or N=3). It isn't helpful to describe low numbers of repeats.

The results are interesting. But, I am not convinced sufficiently novel for this journal. Although it would probably not change the conclusions, I wasn't delighted that so few animals were used in the study. Overall, I think this would be better suited to another journal.

The last 3 paragraphs before the Materials and Methods should consolidate the results of this study and focus and not discuss chicken gut and try to reconcile mechanisms in other species.

Line 20, page 1: "Recent studies have.." does not read well.

Reviewer #2:

Remarks to the Author:

In this article (short communication?), Chevalier and colleagues focus on an interesting and poorly understood phenomenon – here referred to as an isotropic-to-nematic phase transition – that underlies massive structural reorganization of ENS ganglia along the circumferential axis of the developing gut. The authors show that the change in ganglia orientation occurs at different time points depending of the gut segment along the anterior-posterior axis, being first observed in the midgut and later in hindgut and cecum. Other observations suggest that these changes are somehow linked to smooth muscle differentiation, which appears to proceed in a similar spatiotemporal manner a day earlier.

Using *Itgb1*-null mice and SHG microscopy, the authors then conclude that global reorientation of ENS ganglia is the result of beta1-integrin-mediated interactions between ENS cells and smooth muscle-associated collagen type 1.

Although both the topic and the proposed model are interesting, many authors' claims are not well supported by the data shown. Over-interpretation is a major problem here. My specific comments are listed below in accordance with their order of appearance in the manuscript:

1. More detailed explanations are needed for the ImageJ plug-in used to evaluate orientation. For example, how come there is no vertical yellow bars in TuJ1 stained tissues of Fig1 despite the visible presence of many vertical nerve tracts? Accordingly, claims like "interganglionic fibers were always longitudinally oriented" on line 59 should be avoided.
2. Images from the same series should be displayed at same magnification in order to facilitate comparison (e.g., Fig.1b).
3. The text should be clear about how the timing of smooth muscle differentiation was determined (lines 62-63). In this regard, this reviewer is not convinced that the indirect approach used here is appropriate; contractility is the results of interactions between multiple cell types (smooth muscle, ICCs and enteric neurons). SMA staining as displayed in Fig.S2b would be more appropriate.
4. Any idea why the reorganization phenomenon is so poorly effective *ex vivo*?
5. More details are needed about the mouse lines used to obtain *Itgb1*-cKO tissues that can be stained with X-Gal. Is *Rosa26-lacZ* involved?
6. The claim that difference in orientation of *Itgb1*-null ganglia "is more conspicuous at stage P1" (line 84) is not supported by the pictures shown in Fig.2f, which show smaller ganglia in mutant but with grossly similar orientation.
7. Some references are not properly formatted (e.g., line 88)
8. The most important concern is about SHG microscopy data and the conclusion that the reorganization phenomenon is directly controlled by the orientation of collagen 1 fibers. Without independent validation (e.g., immunostaining with specific antibodies or use of knockout tissues), the SHG signal cannot be solely attributed to Col1. Other ECM proteins and even smooth muscle cell proteins (e.g., actin-myosin complexes) could also potentially be detected with this non-specific imaging approach. Why is collagen 6 mentioned and then excluded from the equation? It might play a role even if not permissive to migration. In fact, reorganization could well be due to a balance between permissive and non-permissive molecules.
9. Colors of fig.3 should be displayed separately in addition to the merge images.
10. More detailed explanations are needed to explain the species- and region-specific differences in ENS ganglia orientation. In particular, why are ENS ganglia not circumferentially aligned in chick and human colon despite the presence of circular Col1 fibers (which are considered by the authors as being critical for the reorganization phenomenon)?
11. Hypothesis like this one should be developed further: "Differences in the balance of forces pushing for orientation (ECM-ganglion interactions) and the ones opposing it (cell-cell adhesion and interganglionic fiber tension) are the likely reason of these species-specific variations in the final ENS morphology."
12. The final conclusion should be toned down as the role played by the ECM has not been formally

tested in this study: "...our research stresses the key role played by the extra-cellular matrix in driving major structural rearrangements of the enteric nervous system during mammalian fetal development."

13. It is incorrect to claim that "antibody staining of collagen is always performed on physical tissue sections..." Many different types of collagen can be visualized using whole-mount immunostaining.

Reviewer #3:

Remarks to the Author:

1. The novelty of this paper appears to hinge on the conceptualization and visualization of the changing organization of the developing ENS. Terms such as isotropic, nematic, anisotropic, orientation, and coherence are not well defined. This field is somewhat niche, but could have significance to a wide audience who may not already be familiar with these terms. The authors should clearly explain the significance of orientation and coherence (including orientation vs coherence) with respect to ENS organization. A schematic could be useful.

2. It is not clear which mouse lines were used by the text of the manuscript and methods section. Several of the references cited do not appear to describe the corresponding mouse line.

3. In measuring cell orientation, do the cellular projections (which likely contain the cytoplasmic reporter) account for most changes in cell orientation? If so, is it possible that the cell body remains unchanged but that, by extending projections in a preferential direction, its orientation is perceived to have change by the imaging software? In other words, does "orientation" refer to the cell body or the cell body plus all of its projections?

4. It would be helpful to this reviewer if the authors clearly identified data that reveal a novel finding that has not been previously described elsewhere. Some findings (such as the 2 layers of MP and dependence of ENCC migration on ECM) have already been published.

5. The data relating to motility (Fig S2) seems a bit disjointed, and it's not clear how this contributes to the thesis of this paper.

6. In the *Itgb1*-cKO experiment, what specifically was used as the control?

7. In figure 2E, what determined whether 2, 3, 4, or 5 areas were analyzed per embryo? This difference of sampling could introduce bias into the analysis. Also, the midgut covers a relatively large area, so it would best to sample consistent parts to ensure reproducibility and reduce sampling error. For example, sampling the proximal duodenum and the distal ileum would be a simple way to sample consistently. Lastly, please provide the p values and rationale for why Mann-Whitney test was used (not normally distributed?).

8. The relationship of the claims pertaining to integrins/collagen and CSM need to be further explained. How do these relate to one another with respect to ENS development?

9. Where is the data for this claim?: "Each plane of the MP eventually co-aligns with the circular or longitudinal smooth muscle layer it is in contact with, via smooth-muscle associated ECM".

10. Given lack of reorientation of enteric ganglia in other species, what functional impact does ENCC reorientation play in mammals? Perhaps your motility assay could be applied to this question.

We would like to thank the reviewers for their feedback on our manuscript. You will find a point-by-point response to the criticism raised in green, as well as modifications brought to the manuscript in italic.

Reviewer #1 (Remarks to the Author):

There has been much interest in the colonisation of the embryonic gut by neural crest cells.

In this study by Chevalier et the authors suggest they have revealed the structural arrangement of the myenteric plexus is a result of a dynamic neural crest phase transition (isotropic-to-nematic) that occurs in the fetal gut. The authors suggest that the orientation parameter is critically dependent upon circular smooth muscle associated collagen microfibrils that enteric neural crest cells anchor to via integrin mechanoreceptors. The authors used mouse embryos that expressed YFP or td Tomato in the cytoplasm of all enteric neural crest cells and then used confocal microscopy to follow the morphology of the enteric nervous system from E14.5 to E19.5 in the mid gut, hindgut and cecum. The authors assessed the relative orientation intensity or coherence.

The paper is very descriptive and lacks quantitation in areas. The authors use a knockout mouse (*Itgb1*-CKO) and state they “..revealed more longitudinal and scattered orientations, with lower coherence (~ 0.35)..” This is an imprecise description. And, measurements should be precise with accurate SEMs, not ~ 0.35 .

We have specified the precise averages and standard deviations now, l.131:

*“Control ENCC networks exhibited a circumferential orientation ($80.9 \pm 2.6^\circ$, all results expressed as $\pm SD$) with a high coherence (0.48 ± 0.05). In contrast, the *Itgb1*-cKO network revealed more longitudinal and scattered orientations ($61.7 \pm 10.6^\circ$), with a lower coherence (0.30 ± 0.08).”*

The notion that Integrins are important for enteric neural crest migration is not new. Work (cited in this article) by Breau MA et al. (2006) *Development* 133: 1725-34 has demonstrated the lack of $\beta 1$ integrins leads to a Hirschsprung’s like phenotype.

The authors use the term “integrin mechanoreceptors”. Why ? What is the evidence they perform this role in this environment ?

Beta-integrins are known to mediate mechanical signals between the extracellular matrix and the cell (see for instance Baker & Zaman, *J Biomech.* 2010 Jan 5; 43(1): 38), but since we did not develop this aspect in the manuscript and to alleviate any confusion we changed the term to “integrin receptor”.

The manuscript is quite descriptive. I appreciate that the movement of the ENCC is not easy to describe, but phrases like “..YFP+ cells gradually thicken, coalesce and elongate along the circumferential direction while longitudinal connections between adjacent cell clusters are broken down, resulting in the emergence of rib-like structures” are not very helpful.

We wanted to give a qualitative assessment of the phenomena taking place in VideoS1.

In many places, there are a low number of animals used (often $N=1$, $N=2$ or $N=3$). It isn’t helpful to describe low numbers of repeats.

We added new data so that we now have n=4 at least.

The results are interesting. But, I am not convinced sufficiently novel for this journal. Although it would probably not change the conclusions, I wasn't delighted that so few animals were used in the study. Overall, I think this would be better suited to another journal.

The last 3 paragraphs before the Materials and Methods should consolidate the results of this study and focus and not discuss chicken gut and try to reconcile mechanisms in other species.

Following this comment and those of other reviewers, we have added new data to consolidate our study on the differentiation of smooth muscle in the gut (new Fig.2), in the section on SHG (Fig4&S2), and have expanded the discussion relative to the ECM molecules that can trigger the reorientation.

Line 20, page 1: "Recent studies have.." does not read well.

We have re-phrased this sentence to make it clearer, l.37:

"Recent studies have for instance revealed when different subtypes of ENS neurons differentiate³, how the ENS connects to target cells in the gut⁴ and how the submucosal plexus is formed⁵ and connects to the extrinsic innervation⁵. Several investigations have unraveled how the ENS starts controlling motility fetal intestine^{6,7}. The morphogenesis of ENS ganglia has been the object of particular interest."

Reviewer #2 (Remarks to the Author):

In this article (short communication?), Chevalier and colleagues focus on an interesting and poorly understood phenomenon – here referred to as an isotropic-to-nematic phase transition – that underlies massive structural reorganization of ENS ganglia along the circumferential axis of the developing gut. The authors show that the change in ganglia orientation occurs at different time points depending of the gut segment along the anterior-posterior axis, being first observed in the midgut and later in hindgut and cecum. Other observations suggest that these changes are somehow linked to smooth muscle differentiation, which appears to proceed in a similar spatiotemporal manner a day earlier. Using *Itgb1*-null mice and SHG microscopy, the authors then conclude that global reorientation of ENS ganglia is the result of beta1-integrin-mediated interactions between ENS cells and smooth muscle-associated collagen type 1.

Although both the topic and the proposed model are interesting, many authors' claims are not well supported by the data shown. Over-interpretation is a major problem here. My specific comments are listed below in accordance with their order of appearance in the manuscript:

1. More detailed explanations are needed for the ImageJ plug-in used to evaluate orientation. For example, how come there is no vertical yellow bars in TuJ1 stained tissues of Fig1 despite the visible presence of many vertical nerve tracts? Accordingly, claims like “interganglionic fibers were always longitudinally oriented” on line 59 should be avoided.

The OrientationJ plugin requires an object structure size σ (described in the Materials & Methods). To detect the orientation of ganglia, this size should be on the order of the pixel size of the ganglia, or else it will trace vectors that outline the edge of each ganglion, and are not representative of their average orientation. This also means that some smoothing out is performed and only the dominant orientation of fibers is measured, and not that of all individual fiber tracts; this is what happens in Fig.1a-Tuj. We explain this in more detail in the text in the revised version:

I.56

“We quantitatively assessed cell orientation with the ImageJ “OrientationJ” plugin¹⁷, which yields the average angle of YFP+ structures (0°: longitudinal, 90°: circumferential), and the orientation intensity or coherence (Materials and Methods) at the scale of the ganglion size σ (see Materials and Methods). A high coherence indicates a high degree of anisotropy, whereas low coherence indicates isotropy. Fig.1a shows an example of orientation fields (yellow bars) in E15.5 midgut. The resulting orientation vectors are those of the dominant structure at the scale of the ganglion size, smoothing out smaller structures.”

Materials and methods, I.337:

“We used the Gaussian gradient method and a size σ close to that of the ganglia; choosing too small σ traces the contour of individual ganglia and does not give meaningful information about their average orientation.”

We corrected I.84 to nuance the interpretation: “While the YFP+ structure was circumferentially oriented after the nematic transition, interganglionic fibers were predominantly longitudinally oriented at all stages examined (E14.5-E16.5).”

2. Images from the same series should be displayed at same magnification in order to facilitate comparison (e.g., Fig.1b).

We corrected this in the new Fig.1.

3. The text should be clear about how the timing of smooth muscle differentiation was determined (lines 62-63). In this regard, this reviewer is not convinced that the indirect approach used here is appropriate; contractility is the results of interactions between multiple cell types (smooth muscle, ICCs and enteric neurons). SMA staining as displayed in Fig.S2b would be more appropriate.

We added new data in Figure 2 presenting both SMA whole-mount IHC and the motility data (which we have obtained at E15.5); both are consistent, ie, the onset of motility coincides with the differentiation of smooth muscle in each region of the gut (E13.5 in midgut, E14.5 in hindgut and E15.5-E16.5 in cecum).

4. Any idea why the reorganization phenomenon is so poorly effective ex vivo?

One hypothesis is that the medium we use might not be the optimal one; other investigators have reported using BGJb to grow embryonic guts in culture (Duh. et al., Pediatr. Res. 2000), which might be something to try out in the future.

5. More details are needed about the mouse lines used to obtain Itgb1-cKO tissues that can be stained with X-Gal. Is Rosa26-lacZ involved?

We added details on the mouse lines in the Mat & Methods, I.261:

Mouse models used in this study are the following. The Cre reporter mice Gt(ROSA)26Sortm1(EYFP)Cos⁴⁴, and B6.Cg-Gt(ROSA)26Sortm9(CAG-tdTomato)Hze/J⁴⁵ were referred as YFP^{fl/fl} and Tomato^{fl/fl}, respectively. A transgenic mouse line in which the transgene is under the control of the 3-kb fragment of the human tissue plasminogen activator (Ht-PA) promoter Tg(PLAT-cre)116Sdu¹⁶, was referred as Ht-PA::Cre. YFP^{fl/fl} or Tomato^{fl/fl} were cross with Ht-PA::Cre to generate embryos carrying the reporter protein in migrating NCC and their derivatives.

The mice carrying floxed Itgb1 allele Itgb1tm1⁴⁶ and the mice carrying one Itgb1 null allele Itgb1tm2⁴⁷ were referred as beta1^{neo/fl} and beta1^{neo/+}, respectively. Ht-PA::Cre were crossed with beta1^{neo/+} to generate Ht-PA::Cre; beta1^{neo/+}, which were then crossed with beta1^{fl/fl} to generate embryos with Ht-PA::Cre; beta1^{neo/fl} or Ht-PA::Cre; beta1^{+/fl} genotype as previously described¹⁷. Ht-PA::Cre; beta1^{fl/+} embryos that are heterozygous for Itgb1 in migrating NCC were referred to as controls because Itgb1 heterozygosity has no effect on mice phenotype⁴⁷. Ht-PA::Cre; beta1^{neo/fl} embryos that carry the conditional mutation for Itgb1 in migrating NCC were referred to as Itgb1-cKO. In both controls and Itgb1-cKOs, Cre recombinase-

mediated deletion of the beta1^{fl} allele leads to the expression of beta-galactosidase under the control of the Itgb1 promoter⁴⁶. This allowed us to visualize the NCC and their derivatives including ENS using Xgal staining. The day of the vaginal plug was considered E0.5.

Is Rosa26-lacZ involved?

No. The beta-galactosidase expression in the ENS results from a modification at the Itgb1 locus which produces upon Cre recombinase-mediated deletion of this floxed allele a recombination to give an intact lacZ reporter gene under the control of the Itgb1 promoter described in Potocnik, A. J., Brakebusch, C. & Fässler, R. Fetal and adult hematopoietic stem cells require β 1 integrin function for colonizing fetal liver, spleen, and bone marrow. *Immunity* (2000). doi:10.1016/S1074-7613(00)80216-2 (ref 46)

6. The claim that difference in orientation of Itgb1-null ganglia “is more conspicuous at stage P1” (line 84) is not supported by the pictures shown in Fig.2f, which show smaller ganglia in mutant but with grossly similar orientation.

It is quite difficult to infer an orientation from these very “dot-like” ganglia, although we do understand your comment. Indeed, Itgb1 conditional mutation in NCCs lead to lethality of all the mutants within a month after birth but some newborn mutants dying earlier. In this case some mutants analyzed at P1 exhibit stronger alterations of the ENS than others that render more difficult the analysis of ganglia orientation. We decided to remove this data from the figure and rather refer to images published by Breau et al. 2006 where this is more clearly seen.

7. Some references are not properly formatted (e.g., line 88)

We have corrected this

8. The most important concern is about SHG microscopy data and the conclusion that the reorganization phenomenon is directly controlled by the orientation of collagen 1 fibers. Without independent validation (e.g., immunostaining with specific antibodies or use of knockout tissues), the SHG signal cannot be solely attributed to Col1. Other ECM proteins and even smooth muscle cell proteins (e.g., actin-myosin complexes) could also potentially be detected with this non-specific imaging approach. Why is collagen 6 mentioned and then excluded from the equation? It might play a role even if not permissive to migration. In fact, reorganization could well be due to a balance between permissive and non-permissive molecules.

There are several points in your comment: 1) what molecules does SHG reveal ? 2) what molecules are responsible for the orientation of the ENCCs ?

Concerning the first question, fibrillar collagen I is the main contributor to SHG signal. Fibrillar collagen III and V are often found associated with collagen I (6, 7), but they do not give SHG contrast. These three fibrous collagen types are the main ones found in the intestine (respectively 68, 20 and 12 % in the human gut(3)). Co-labeling of collagen I and SHG have been performed and the two signals overlap in liver samples(2). Our results are consistent with the collagen I immunostainings on E18.5 mouse gut performed by Ormestad et al.(4): they find collagen I above, in and below the smooth muscle actin positive muscle layer (see Fig.3H-J of their paper). The strongest SHG signal we obtain is in the submucosa (below the

SMA layer), consistent with their (4) and other findings (3). Actomyosin can also yield SHG signal, however it is much weaker than that of collagen (1). To ascertain that the signal we recover does not come from actomyosin, we have performed SMA whole-mount immunostaining.

The two signals are very different, especially because the first layer that is observed in the SMA stained samples at E17.5 is the longitudinal muscle layer – in contrast in the SHG images we do not observe any longitudinal signal in this outer layer in the duodenum. The SMA signal is bundled in much thicker fibers than the fibers observed by SHG. Finally, the most intense signal is retrieved from SHG at the submucosa, which is not the case of alpha-actin which is concentrated in the smooth muscle. All in all we therefore do not think that actomyosin could be confused with the signal emanating from collagen I fibers. Note also that smooth muscle cells are known to produce collagen I extracellular matrix in the gut (5).

The second question concerning what ECM molecule(s) trigger(s) ENCC reorientation is more difficult to answer unambiguously on the basis of the data we have at hand. A good candidate molecule would 1) be present in the embryonic gut at the level of the myenteric plexus at the stages investigated (E14.5-E19.5), 2) be permissive to ENCC adhesion, 3) present a circumferential organization, which means be of fibrous nature, 4) be associated / produced by the smooth muscle, because we show that there is a kinetic correlation between the appearance of smooth muscle and the reorientation process. We have demonstrated that collagen I presents all of these characteristics and is therefore a suitable candidate. Collagen III and V are associated with collagen I and therefore it is not meaningful to discuss their contribution separately from that of collagen I. Collagen IV is not fibrous, and is absent from the smooth muscle layer and the myenteric plexus in the colon (see Fig.2 of Ormestad et al.(4)), which disqualifies it because we also observe reorientation in the colon. Collagen VI is a repellent of neural crest: overexpression has been found to cause a Hirschsprung type phenotype. Video S1 shows that the spaces devoid of ENCCs appear because intercellular adherence attracts the ENCCs around each elongating ganglion, locally depleting the surrounding ENCCs, and thus creating a void; we therefore think it unnecessary to invoke spatially periodic expression of an ECM molecule that would actively repel the ENCCs from these interganglionic spaces. We cannot however exclude that other non-collagenous ECM molecules like fibronectin, who is a major driver of ENCC migration, could be produced by the smooth muscle and trigger the orientation.

1. **Campagnola PJ, Millard AC, Terasaki M, Hoppe PE, Malone CJ, Mohler WA.** Three-dimensional high-resolution second-harmonic generation imaging of endogenous structural proteins in biological tissues. *Biophys J* 82: 493–508, 2002. doi: 10.1016/S0006-3495(02)75414-3.

2. **Gailhouste L, Grand Y Le, Odin C, Guyader D, Turlin B, Ezan F, Désille Y, Guilbert T, Bessard A, Frémin C, Theret N, Baffet G.** Fibrillar collagen scoring by second harmonic microscopy: A new tool in the assessment of liver fibrosis. .
3. **Graham MF, Diegelmann RF, Elson CO, Lindblad WJ, Gotschalk N, Gay S, Gay R.** Collagen content and types in the intestinal strictures of Crohn's disease. *Gastroenterology* 94: 257–265, 1988. doi: 10.1016/0016-5085(88)90411-8.
4. **Ormestad M, Astorga J, Landgren H, Wang T, Johansson BR, Miura N, Carlsson P.** Foxf1 and Foxf2 control murine gut development by limiting mesenchymal Wnt signaling and promoting extracellular matrix production. *Development* 133: 833–843, 2006. doi: 10.1242/dev.02252.
5. **Perr HA, Grider JR, Mills AS, Kornstein M, Turner DA, Diegelmann RF, Graham MF.** Collagen production by human smooth muscle cells isolated during intestinal organogenesis. .
6. **Ranjit S, Dvornikov A, Stakic M, Hong SH, Levi M, Evans RM, Gratton E.** Imaging Fibrosis and Separating Collagens using Second Harmonic Generation and Phasor Approach to Fluorescence Lifetime Imaging. .
7. **Wang C, Brisson BK, Terajima M, Li Q, Hoxha K, Han B, Goldberg AM, Sherry Liu X, Marcolongo MS, Enomoto-Iwamoto M, Yamauchi M, Volk SW, Han L.** Type III collagen is a key regulator of the collagen fibrillar structure and biomechanics of articular cartilage and meniscus. .

We have summarized these considerations in the revised manuscript as such:

Results section, I.169:

“The collagen distribution in the MP, SMC and submucosa we report is consistent with immunostaining of collagen I on E18.5 duodenum thin-sections presented by other investigators”.

Discussion section, I.197:

“Candidate ECM molecules that could serve as a template for ENCC orientation via β 1-integrins should fulfill the following criteria: 1) be present in the embryonic gut at the level of the myenteric plexus at the stages investigated (E14.5-E19.5), 2) be permissive to ENCC adhesion, 3) present a circumferential organization, that is be of fibrous nature, 4) be associated / produced by the smooth muscle, because we showed that there is a kinetic correlation between the appearance of smooth muscle and the reorientation process.

Collagen I, III and V are the main collagen molecules present in the gut, representing respectively 68, 20 and 12 % of collagens in the human intestine²⁴. Collagen III and V are generally found associated with collagen I fibers^{25,26} and for the sake of this discussion we will therefore consider these three fibrous collagen types as one entity. Collagen I have been shown to be permissive to ENCC migration²²; in particular, the orientation and translocation of neural crest cells along (parallel to) collagen I fibers has been reported by Davis²⁷. Importantly, collagen I, III and V have been shown to be produced by fetal human intestinal smooth muscle cells²⁸. We have shown here by second harmonic generation microscopy that collagen I is present at the right location (myenteric plexus), stage (E14.5-E17.5) and with the proper circumferential orientation. In contrast, collagen IV which forms a sheet-like structure, although it also is permissive to migration²², is not present at the level of the myenteric plexus in the hindgut when the orientation transition occurs²³. Type VI collagen is mostly localized at the basal membrane of the epithelium²⁹, but is also secreted by ENCCs. It however was found to have an inhibitory effect on migration, with overexpression triggering a Hirschsprung type phenotype³⁰. Collagen XVIII is only secreted by ENCCs at the colonization wavefront³¹. We can therefore conclude that, among collagens, collagen I fibers (and associated type III and V) fulfills the criteria of an ideal template ECM for ENCC orientation. We cannot however exclude that other, non-

collagenous ECM molecules that support ENCC migration, like fibronectin³⁰, could be produced in an oriented way by the CSM and play a role in driving the orientation transition of ENCCs.”

Materials and Methods section, I.329:

“SHG allows for quantitative evaluation of collagen fiber density. SHG is sensitive to collagen type I and type II⁴⁸, but since only type I is found in the gut (where it is the main collagen type expressed²⁴), we image here the collagen I distribution. Comparative images of collagen I distribution by immunostaining and of SHG generation in liver samples have for example shown almost complete overlap⁴⁹. We note that SHG is insensitive to non-fibrillar collagen like type IV; type III does not generate a measurable SHG signal⁴⁸.”

9. Colors of fig.3 should be displayed separately in addition to the merge images.

We have modified Fig.3 (new Fig. 4 &S2) to show the signals from SHG and ENCCs separately, and as merge images.

10. More detailed explanations are needed to explain the species- and region-specific differences in ENS ganglia orientation. In particular, why are ENS ganglia not circumferentially aligned in chick and human colon despite the presence of circular Col1 fibers (which are considered by the authors as being critical for the reorganization phenomenon)?

11. Hypothesis like this one should be developed further: “Differences in the balance of forces pushing for orientation (ECM-ganglion interactions) and the ones opposing it (cell-cell adhesion and interganglionic fiber tension) are the likely reason of these species-specific variations in the final ENS morphology.”

We added the following to clarify our thought on interganglionic fiber tension, I.236:

“Interganglionic fibers in chicken are noticeably thicker than in mice and longitudinally oriented; the pulling force they exert on ganglia may offset the tendency of ganglia to orient circumferentially with the underlying ECM”

12. The final conclusion should be toned down as the role played by the ECM has not been formally tested in this study: “...our research stresses the key role played by the extra-cellular matrix in driving major structural rearrangements of the enteric nervous system during mammalian fetal development.”

We have rephrased this as “our research stresses the key role played by interactions between enteric neural crest cells and the extra-cellular matrix“ to more accurately reflect the *Itgb1cKO* model we use in our report.

13. It is incorrect to claim that “antibody staining of collagen is always performed on physical tissue sections...” Many different types of collagen can be visualized using whole-mount immunostaining.

We have modified this claim to “most often performed on tissue section”.

Reviewer #3 (Remarks to the Author):

1. The novelty of this paper appears to hinge on the conceptualization and visualization of the changing organization of the developing ENS. Terms such as isotropic, nematic, anisotropic, orientation, and coherence are not well defined. This field is somewhat niche, but could have significance to a wide audience who may not already be familiar with these terms. The authors should clearly explain the significance of orientation and coherence (including orientation vs coherence) with respect to ENS organization. A schematic could be useful.

We strived to give more detailed explanation for isotropy, anisotropy and their relation to coherence at the beginning of the results section in the revised version, l.59:

“A high coherence indicates a high degree of anisotropy, i.e., that they exhibit a distinct preferential direction. In contrast, a low coherence indicates isotropy, i.e., that the structures are only weakly or not oriented along any direction.”

We also detail the use of the terms nematic at the beginning for the results section, l.184:

“In analogy to the physics of liquid crystals, the orientation transition can be assimilated to an isotropic-to-nematic transition, where the isotropic phase does not exhibit a preferential direction, whereas the nematic phase is ordered along one direction (the circumference of the gut).”

2. It is not clear which mouse lines were used by the text of the manuscript and methods section. Several of the references cited do not appear to describe the corresponding mouse line.

We have now outlined in more detail the mouse lines in the Materials & Methods section, l.264:

“Mouse models used in this study are the following. The Cre reporter mice Gt(ROSA)26Sortm1(EYFP)Cos 44, and B6.Cg-Gt(ROSA)26Sortm9(CAG-tdTomato)Hze/J 45 were referred as YFPfl/fl and Tomatofl/fl, respectively. A transgenic mouse line in which the transgene is under the control of the 3-kb fragment of the human tissue plasminogen activator (Ht-PA) promoter Tg(PLAT-cre)116Sdu 16, was referred as Ht-PA::Cre. YFPfl/fl or Tomatofl/fl were cross with Ht-PA::Cre to generate embryos carrying the reporter protein in migrating NCC and their derivatives.

The mice carrying floxed Itgb1 allele Itgb1tm1 46 and the mice carrying one Itgb1 null allele Itgb1tm2 47 were referred as beta1fl/fl and beta1neo/+, respectively. Ht-PA::Cre were crossed with beta1neo/+ to generate Ht-PA::Cre; beta1neo/+, which were then crossed with beta1fl/fl to generate embryos with Ht-PA::Cre; beta1neo/fl or Ht-PA::Cre; beta1+/fl genotype as previously described 17. Ht-PA::Cre; beta1fl/+ embryos that are heterozygous for Itgb1 in migrating NCC were referred to as controls because Itgb1 heterozygosity has no effect on mice phenotype 47. Ht-PA::Cre; beta1neo/fl embryos that carry the conditional mutation for Itgb1 in migrating NCC were referred to as Itgb1-cKO. In both controls and Itgb1-cKOs, Cre

recombinase-mediated deletion of the beta1fl allele leads to the expression of beta-galactosidase under the control of the Itgb1 promoter 46. This allowed us to visualize the NCC and their derivatives including ENS using Xgal staining. The day of the vaginal plug was considered E0.5.”

3. In measuring cell orientation, do the cellular projections (which likely contain the cytoplasmic reporter) account for most changes in cell orientation? If so, is it possible that the cell body remains unchanged but that, by extending projections in a preferential direction, its orientation is perceived to have change by the imaging software? In other words, does “orientation” refer to the cell body or the cell body plus all of its projections?

The cell bodies cluster in the longitudinal direction, forming an elongated ganglia, whereas interganglionic fibers run longitudinally. Individual cell bodies are not circumferentially oriented. This is distinctly seen on the new images in Fig.1d below obtained by Tuj1 staining on E18.5 gut (duodenum), resolved with a better microscope which allows to see the individual cell bodies in each ganglion. We have added this information to the manuscript, l.88:

“This was confirmed by high resolution confocal microscopy (using a Zeiss CSU-W1 microscope) of Tuj1+ immunolabeled whole mounts (Fig.1d). The higher resolution of this microscope also allowed us to reveal the cellular structure of each ganglion: individual cell bodies are not circumferentially oriented, but the ganglion as the aggregate of these cell bodies is (Fig.1d).”

4. It would be helpful to this reviewer if the authors clearly identified data that reveal a novel finding that has not been previously described elsewhere. Some findings (such as the 2 layers of MP and dependence of ENCC migration on ECM) have already been published.

The novel findings in this report are the following:

- We provide the first live imaging of the orientation phenomenon (VideoS1) and developed methods to quantify the orientation.
- We find a kinetic correlation between ENS orientation and the differentiation of smooth muscle
- We characterize how much ganglia are disoriented in ENCC-ECM beta1-integrin KO mice; while a previous manuscript on this mouse focused on the effect of this mutation on ENCC migration, it did not investigate the effects on the orientation of the ENS in ganglionic parts of the gut
- We obtain the first 3D images of fibrous collagen in contact with the ENCCs showing that collagen I could be a suitable ECM molecule driving the reorientation

We have summarized these points both in the abstract and in a summary paragraph at the beginning of the discussion section in the revised version.

5. The data relating to motility (Fig S2) seems a bit disjointed, and it’s not clear how this contributes to the thesis of this paper.

We now provide a new figure 2 showing both smooth-muscle actin wholemount IHC and the motility data (which we have completed at E15.5 also); we used motility as a proxy to

determine when circular smooth muscle differentiates in the gut, now we show that the two are identical (ie, the time of onset of motility coincides with time of appearance of alpha-SMA).

6. In the *Itgb1*-cKO experiment, what specifically was used as the control?

We specify the control in the Materials and Methods now, l.275:

“Ht-PA::Cre; beta1fl/+ that are heterozygous for Itgb1 in migrating NCC were referred to as controls because Itgb1 heterozygosity has no effect on mice phenotype (Fassler and Meyer, 1995; Stephens et al., 1995).”

7. In figure 2E, what determined whether 2, 3, 4, or 5 areas were analyzed per embryo? This difference of sampling could introduce bias into the analysis. Also, the midgut covers a relatively large area, so it would best to sample consistent parts to ensure reproducibility and reduce sampling error. For example, sampling the proximal duodenum and the distal ileum would be a simple way to sample consistently. Lastly, please provide the p values and rationale for why Mann-Whitney test was used (not normally distributed?).

We sampled all regions that could be analyzed (in focus, not too curved segment) we could with the images we had at hand. Unfortunately we do not have the separation of the different regions for duodenum, jejunum, ileum, and have therefore pooled them as “midgut”. The rationale for the Mann-Whitney is indeed that the data is not normally distributed (we do not have enough samples to prove normality).

8. The relationship of the claims pertaining to integrins/collagen and CSM need to be further explained. How do these relate to one another with respect to ENS development?

We explain this in more detail in the discussion section now:

“What ECM molecule triggers the orientation transition ?

Candidate ECM molecules that could serve as a template for ENCC orientation via β 1-integrins should fulfill the following criteria: 1) be present in the embryonic gut at the level of the myenteric plexus at the stages investigated (E14.5-E19.5), 2) be permissive to ENCC adhesion, 3) present a circumferential organization, that is be of fibrous nature, 4) be associated / produced by the smooth muscle, because we showed that there is a kinetic correlation between the appearance of smooth muscle and the reorientation process.

Collagen I, III and V are the main collagen molecules present in the gut, representing respectively 68, 20 and 12 % of collagens in the human intestine²⁵. Collagen III and V are generally found associated with collagen I fibers^{25,26} and for the sake of this discussion we will therefore consider these three fibrous collagen types as one entity. Collagen I have been shown to be permissive to ENCC migration^{22,23} ; in particular, the orientation and translocation of neural crest cells along (parallel to) collagen I fibers has been reported by Davis²⁸. Importantly, collagen I, III and V have been shown to be produced by fetal human intestinal smooth muscle cells²⁹. We have shown here by second harmonic generation microscopy that collagen I is present at the right location (myenteric plexus), stage (E14.5-E17.5) and with the proper circumferential orientation. In contrast, collagen IV which forms a sheet-like structure, although it also is permissive to migration²², is not present at the level of

the myenteric plexus in the hindgut when the orientation transition occurs²³. Type VI collagen is mostly localized at the basal membrane of the epithelium³⁰, but is also secreted by ENCCs. It however was found to have an inhibitory effect on migration, with overexpression triggering a Hirschsprung type phenotype³¹. Collagen XVIII is only secreted by ENCCs at the colonization wavefront³². We can therefore conclude that, among collagens, collagen I fibers (and associated type III and V) fulfills the criteria of an ideal template ECM for ENCC orientation. We cannot however exclude that other, non-collagenous ECM molecules that support ENCC migration, like fibronectin³¹, could be produced in an oriented way by the CSM and play a role in driving the orientation transition of ENCCs.”

9. Where is the data for this claim?: “Each plane of the MP eventually co-aligns with the circular or longitudinal smooth muscle layer it is in contact with, via smooth-muscle associated ECM”.

We have removed this statement because it is true that longitudinal orientation of interganglionic fibers is observed already at E15.5, before the emergence of the longitudinal smooth muscle layer (E16.5).

10. Given lack of reorientation of enteric ganglia in other species, what functional impact does ENCC reorientation play in mammals? Perhaps your motility assay could be applied to this question.

This is a very interesting question which we are investigating and that will be the object of a separate report; we are seeing that the deformation patterns of the ganglia in response to mechanical stress (from a bolus, or from spontaneous muscle contractions) are very different in these two species. We have added the following sentence in the discussion section to open up our investigation to this question, l.238:

“An interesting question is the implication of these differences of ENS geometry on the physiology of the intestine. Neurons in the ENS are known to be mechanosensitive³⁵ and we expect that the mechanical stresses (due to the presence of food bolus or spontaneous muscle contractions) experienced by circumferential vs hexagonal ENS lattices will be significantly different.”

REVIEWERS' COMMENTS:

Reviewer #1 (Remarks to the Author):

The authors present compelling new data that in the mammalian fetus, the smooth-muscle associated extracellular matrix plays a key role in the reorientation transition of the enteric nervous system. The authors have done a fine job in this revision. It is a significant improvement. The conclusions appear valid and the manuscript well written, with suitable numbers of replicate experiments. No concerns with statistics and figures look clear.

Reviewer #2 (Remarks to the Author):

Although the revised manuscript has been generally improved, some important concerns remain:

1. Why being vague about the percentage (10-20%) of ex vivo tissues that behaved as in vivo? If $n=10$, then it is either 1 (10%) or 2 (20%).

2. The interpretation of SHG microscopy data is still problematic. Without independent validation in the same type (developing gut) of tissues (e.g., immunostaining with specific antibodies or use of knockout tissues), it is hard to trust that the SHG signal is solely due to Col1 as claimed by the authors (bottom of p.9). This group can perform whole-mount staining of developing guts, so why not at least try with a Col1 antibody? Many different types of collagen can be visualized using whole-mount immunostaining. I do not think that referring to a prior SHG study of liver tissues or another study of Col1 tissue distribution at later stages of development is adequate for instilling confidence in the data shown here. The new Fig4 (which now correctly present data for each fluorescent channel separately), clearly shows extensive overlap with myenteric plexus-associated ENCCs (there is virtually no green signal outside ENCCs in the $z=80\mu\text{m}$ panel of Fig4a). Is Col1 expressed and secreted by ENCCs? How can this be reconciled with the proposed mechanism that this is the CSM-associated Col1 that matters? This also raises the concern that there is no data for the period where the reorientation phenomenon actually takes place; data shown are only for earlier and later time points. Moreover, the way data are presented is confusing with a mix of x-y and z-y views; an indication of the axes would be needed ("one gut wall" makes no sense).

Minor comments:

- Introduction (line 42): it is incorrect to suggest that enteric glia are restricted to ganglia periphery.
- Results (line 94): insert "clusters" before "YFP+ cells"
- Results (line 134): insert "mean" before " \pm SD"
- Revise spelling of ENCC/ENCCs and NCC/NCCs

Reviewer #3 (Remarks to the Author):

Thank you, this revised manuscript is significantly improved as it more clearly explicates its methods and findings, and the reviewer comments are satisfactorily addressed.

Dear reviewer,

We thank you for the feedback, please find below the required clarifications.

Reviewer #2 (Remarks to the Author):

Although the revised manuscript has been generally improved, some important concerns remain:

1. Why being vague about the percentage (10-20%) of ex vivo tissues that behaved as in vivo? If n=10, then it is either 1 (10%) or 2 (20%).

Reorientation occurred in all samples but not in all regions along the midgut (for each sample we image 40-50 regions from the rostral to the caudal end of the gut). In this experiment, we assessed orientation of ganglia qualitatively (no, yes, slightly) and the range is related to the uncertainty in classifying in these categories. We now precise this in the manuscript:

“While orientation in-vivo is a robust, systematic phenomenon, we could only observe it in 10-20% of the midgut regions examined (n=10 samples and n=40-50 images along the midgut for each sample) after 2 days in culture (E14.5+2).”

2. The interpretation of SHG microscopy data is still problematic. Without independent validation in the same type (developing gut) of tissues (e.g., immunostaining with specific antibodies or use of knockout tissues), it is hard to trust that the SHG signal is solely due to Col1 as claimed by the authors (bottom of p.9). This group can perform whole-mount staining of developing guts, so why not at least try with a Col1 antibody? Many different types of collagen can be visualized using whole-mount immunostaining. I do not think that referring to a prior SHG study of liver tissues or another study of Col1 tissue distribution at later stages of development is adequate for instilling confidence in the data shown here.

We have now performed collagen 1 antibody staining and present the results in Fig.S3 and Video S5. We have added the following to the manuscript:

I.181

“We finally compared the fiber pattern revealed by SHG microscopy with those labeled by collagen I whole-mount antibody immunohistochemistry on E17.5 gut (Figure S3, Video S5). We found that collagen fibers are wound circularly at the level of the myenteric plexus, as found by SHG in Fig.4b. We could not however reveal collagen deeper in the mucosa, either because the antibody did not penetrate the deeper tissue layers, or because we used a spinning disk microscope that has lower penetration depth than the two-photon SHG method. Antibody staining further labeled individual collagen rich cells below the myenteric plexus, that we did not observe by SHG microscopy.”

Mat. Meth., I.329:

“For whole-mount collagen I (Sigma SAB1402151) IHC, primary antibody was incubated at 1:100 for 3 days, and revealed with Alexa647 secondary antibody (1:400, overnight).”

The new Fig4 (which now correctly present data for each fluorescent channel separately), clearly shows extensive overlap with myenteric plexus-associated ENCCs (there is virtually no green signal outside ENCCs in the z=80um panel of Fig4a). . Is Col1 expressed and

secreted by ENCCs? How can this be reconciled with the proposed mechanism that this is the CSM-associated Col1 that matters?

The merge image in Fig.4a z=80 μm shows the collagen and the ENCCs overlap to a significant extent, but also that the SHG signal is located more inside the gut, i.e., towards the circular smooth muscle, consistent with our hypothesis that the collagen is secreted by the circular smooth muscle, not by the ENCCs (l.209 : *“Importantly, collagen I, III and V have been shown to be produced by fetal human intestinal smooth muscle cells²⁹”*). This layer is very thin at E14.5, because smooth muscle has just differentiated, and becomes thicker at E17.5. Note also that on the Fig.4a z=80 μm the serosa is visible at the outer periphery, consistent with the fact that this membrane is rich in collagen. We have clarified this in the text as such (l.164):

“This fiber layer was very thin (a few μm) at E14.5 (Fig.4a), co-located to a large extent with the ENCCs (Fig.4a z=80 μm , merge), but was situated more towards the inside of the intestine, at the boundary between the circular smooth muscle and the myenteric plexus. SHG microscopy also revealed the collagen-rich serosa (Fig.4a z=80 μm). At E17.5, the collagen fiber layer became much thicker ($\sim 50 \mu\text{m}$, Fig.4b). The intensity of the SHG signal just below the myenteric plexus increased 5-fold between E14.5 and E17.5, from 25 ± 5 pixel units ($n=7$ stacks from $n=4$ guts) to 128 ± 34 ($n=9$ stacks from $n=4$ guts).”

This also raises the concern that there is no data for the period where the reorientation phenomenon actually takes place; data shown are only for earlier and later time points. Moreover, the way data are presented is confusing with a mix of x-y and z-y views; an indication of the axes would be needed (“one gut wall” makes no sense).

We replaced “one gut wall” with “longitudinal optical section of one border of the gut”. The axis are all x-y in Fig.4; we have added information in the longitudinal optical sections to understand where is inside and outside the gut.

Minor comments:

-Introduction (line 42): it is incorrect to suggest that enteric glia are restricted to ganglia periphery.

We have made the statement more accurate:

“Neuronal cell bodies occupy the central part of ganglia while glial cells are present both inside the ganglion, at its outer periphery, and in the interganglionic fiber tracts⁹.”

-Results (line 94): insert “clusters” before “YFP+ cells”

-Results (line 134): insert “mean” before “ \pm SD”

-Revise spelling of ENCC/ENCCs and NCC/NCCs

We have corrected this.